# Larger whole brain grey matter associated with long-term Sahaja Yoga Meditation: A detailed area by area comparison

Sergio Elías Hernández[1]*, Roberto Dorta[2], José Suero[3], Alfonso Barros-Loscertales[4], José Luis González-Mora[5], Katya Rubia[6]

1 Department of Ingeniería Industrial, Universidad de La Laguna, Tenerife, Spain, 2 Department of Matemáticas, Estadística e Investigación Operativa, Universidad de La Laguna, Tenerife, Spain, 3 Centro de Salud Jazmín, Sermas, Madrid, Spain, 4 Department of Psicología Básica, Clínica y Psicobiología, Universitat Jaume I, Castellón, Spain, 5 Department of Fisiología, Universidad de La Laguna, Tenerife, Spain, 6 Institute of Psychiatry, Psychology and Neuroscience, King's College London, London, United Kingdom

* sehdez@ull.edu.es

**Data Availability Statement:** All relevant data are within the paper and its Supporting information files.

## Abstract

### Objectives

Our previous study showed that long-term practitioners of Sahaja Yoga Meditation (SYM) had around 7% larger grey matter volume (GMV) in the whole brain compared with healthy controls; however, when testing individual regions, only 5 small brain areas were statistically different between groups. Under the hypothesis that those results were statistically conservative, with the same dataset, we investigated in more detail the regional differences in GMV associated with the practice of SYM, with a different statistical approach.

### Design

Twenty-three experienced practitioners of SYM and 23 healthy non-meditators matched on age, sex and education level, were scanned using structural magnetic resonance imaging (MRI). Their GMV were extracted and compared using Voxel-Based Morphometry (VBM). Using a novel ad-hoc general linear model, statistical comparisons were made to observe if the GMV differences between meditators and controls were statistically significant.

### Results

In the 16 lobe area subdivisions, GMV was statistically significantly different in 4 out of 16 areas: in right hemispheric temporal and frontal lobes, left frontal lobe and brainstem. In the 116 AAL area subdivisions, GMV difference was statistically significant in 11 areas. The GMV differences were statistically more significant in right hemispheric brain areas.

### Conclusions

The study shows that long-term practice of SYM is associated with larger GMV overall, and with significant differences mainly in temporal and frontal areas of the right hemisphere and

**Funding:** The author(s) received no specific funding for this work.

the brainstem. These neuroplastic changes may reflect emotional and attentional control mechanisms developed with SYM. On the other hand, our statistical ad-hoc method shows that there were more brain areas with statistical significance compared to the traditional methodology which we think is susceptible to conservative Type II errors.

## Introduction

Meditation is a general term that includes a large variety of practices that mainly focus on the inner observation of the body and the mind. The western goal of most meditation techniques is to achieve an improved control of attention and emotions in order to live a more balanced, stress-free and healthier life. On the other hand, yoga includes many different techniques among which meditation (dhayana in classical yoga) has a main role. If we travel back to the origins of yoga, the first known treaty "The yoga sutras of Patanjali" mentions that "Yoga is the suppression of the modifications of the mind" [1, 2]. In ancient yoga, a higher state of consciousness called Nirvichara Samadhi was described, in today's words Nirvichara could be translated as "mental silence" or "thoughtless awareness". In this state, the mind has none thoughts and there is inner calm in a state of inner pure joy and the attention is focused on each present moment. Sahaja Yoga Meditation (SYM) shares the goals of Patanjali's Yoga Sutras to achieve the state of Nirvichara or mental silence.

SYM, presumably through the regular achievement of the state of mental silence, has shown health benefits in disorders that are often associated with recurrent or repetitive negative thoughts, such as: depression, stress, anxiety, and attention-deficit/hyperactivity disorder [2–7]. Other studies on SYM have shown beneficial effects in treating physiological and neurological diseases such as asthma [8], high blood pressure [6], menopause [9] and epilepsy [10–12], for a meta-analysis see [8]. Furthermore, the frequency with which the practitioners perceive the state of mental silence has been shown to be associated with better physical and mental health [13].

Neuroplasticity is one of the most commonly used terms in today's neuroscience to express the capacity of our human brain to change permanently. One of the key insights over the past 2 decades of neuroimaging research has been that the human brain, even in adulthood, is not static, but on the contrary is a dynamic system that has the ability to shape itself. One of the key fascinating questions that researchers try to answer is hence: how can we improve our brain structure and function? One potential non-pharmacological way to shape our brain could be through meditation [14].

Neuroplasticity can be measured by changes in grey matter volume (GMV). Many studies have shown that brain areas that are more utilized through practice of a particular skill for example, in music [15], or high performance sports [16], can become enlarged. It has even been shown that relatively short periods of training of a particular skill, such as 3 months of training to juggle or 3 months of studying for an exam in students can lead to transient changes in the relevant brain areas such as visual-spatial perception regions for juggling [17, 18] or the hippocampus and parietal lobe for memory storage in medical students preparing for an exam [19, 20].

Voxel Based morphometry (VBM) is the most used automated technique to measure GMV by means of MRI scans. In most cases researchers follow the steps provided by the VBM authors of the technique [21–24]. VBM has evolved [21] and the different steps like segmentation and normalization have been improved with each new software version [24, 25].

In most cases, the statistical path followed to compare GMV mean differences between groups has been with ANCOVAs, were typically total intracranial volumes (TIV), sex and age are treated as nuisance covariates. This statistical method is based on random field theory [21, 26]. Another important point to consider is that structural images display local variation in smoothness, which implies that cluster-level corrections should be applied using Random Field Theory and non-stationary correction [27].

In our previous structural MRI study, we showed that 23 long-term practitioners of SYM compared to healthy controls had 6.9% significantly larger GMV in the whole brain [28] which represent, as far as we know, the highest GMV difference shown between groups of healthy volunteers. However, this significant whole brain difference was related with only two relatively small areas showing statistical significance located in right insula and right inferior temporal gyrus with respective volumes of 564 and 739 mm$^3$. Considering the concern of incurring in Type II errors (false negatives or conservative assumptions), the aim of our study was to analyse in more detail how the GMV differences are distributed across the whole brain. This new study is based in two key issues: 1) The development of an statistical ad-hoc general linear model (AH-GLM) that adapts itself on each brain area depending on the significance of covariates of that particular area; and 2) The parcellation of the human brain using 2 different methods i. Based on the human brain lobes: frontal, temporal, etc, that gives rise to 16 different brain areas and ii. Using the more specific automated anatomical labelling (AAL) of 116 brain areas [29, 30]. We used these two brain atlas subdivisions because they are widely used in the neuroscience literature and because the proposed methodology allows us to study both of them in detail. This way we could test the application of our ad-hoc method in 2 different scenarios and observe if the results obtained provided some overlap and coherence.

The key question for this analysis was whether there were any areas that differed between long-term meditators and healthy controls which were overlooked in our previous paper [28] due to a Type II error correction effect.

## Materials and methods

### Participants

Forty-six white Caucasian, right-handed, healthy volunteers, between 21 and 63 years participated in this study. Twenty-three of them were long-term, expert practitioners of SYM (17 females and 6 males) while the other 23 (also 17 females and 6 males) were non-meditators matched on sex, education degree, body mass index and age (see Table 1). All volunteers informed that they had no physical or mental illness, no history of neurological disorders, and no addiction to alcohol, nicotine or drugs.

**Table 1. Demographic characteristics of the groups.**

|  | Meditators Mean (SD) | Controls Mean+ (SD) | t(df = 44) | p-value* |
|---|---|---|---|---|
| Volunteers N˚ | 23 | 23 |  |  |
| Age (years) | 46.5 (11.4) | 46.9 (10.9) | -0.13 | 0.89 |
| Age range (years) | 20.3–63.1 | 21.3–63.3 |  |  |
| Education degree, 0 to 6 | 3.78 (1.2) | 4.04 (1.36) | 0.69 | 0.50 |
| Height (cm) | 167.0 (8.8) | 167.2 (7.6) | 0.09 | 0.93 |
| Weight (Kg) | 69.5 (14.6) | 71.7 (14.5) | 0.53 | 0.60 |
| Body mass index | 24.9 (4.5) | 25.5 (3.9) | 0.54 | 0.60 |

*p-values represent group differences between meditators and controls using two-tailed independent samples t-tests.

Meditators had more than 5 years of daily meditation practice in SYM (mean 14.1 SD (6.1) years); the daily average time dedicated to meditation was 84.7 (32.2) minutes.

Before their participation in this research, all volunteers filled in different questionnaires to validate their individual health status, education and age. Additionally, meditators filled in other questionnaire that asked about their experience in SYM, including: average time dedicated to meditation per day, frequency of the perception of the state of mental silence, total hours of meditation and years of practice of SYM.

All participants signed informed consent to participate freely. This study was approved by the Ethics Committee of the University of La Laguna.

## MRI acquisition

All images were obtained on a 3T MRI Scanner. High resolution sagittally oriented anatomical images were collected. A 3D fast spoiled-gradient recalled pulse sequence was obtained (TR = 8.8 ms, TE = 1.7 ms, flip angle = 10˚, matrix size = 256 × 256 pixels, 1 × 1 mm in plane resolution, spacing between slices = 1 mm plus 0 mm interslice gap, slice thickness = 1 mm). Total acquisition time was 13 minutes.

## Voxel-based morphometry

Voxel-based morphometry (VBM) [21] with DARTEL was conducted using the SPM12 software package (Statistical Parametric Mapping software: http://www.fil.ion.ucl.ac.uk/spm/). Processing steps were performed as suggested by the method's author [31]. VBM with DARTEL has been shown to be more sensitive than standard VBM [24] and provides results comparable to those achieved with manual segmentation [32].

The procedure followed these steps: 1. All T1-weighted anatomical images were displayed to screen to verify they were free from gross anatomical abnormalities. 2. For better registration, the T1 images were manually centred at the anterior commissure and reoriented according to the anterior–posterior commissure line. 3. Using the New Segment procedure in SPM12, images were segmented into: Grey matter (GM), White matter (WM) and Cerebrum Spinal Fluid (CSF), a segmentation that provides acceptable substitute for labour intensive manual estimates [25]. 4. The DARTEL routine inside SPM12 was used to spatially normalize the segmented images [24]. The image intensity of each voxel was modulated by the Jacobian determinants to ensure that regional differences in the total amount of GMV were conserved. 5. The final DARTEL template image modulated in the previous step was registered to MNI space so that all the individual spatially normalised scans were brought into MNI space. 6. Finally, the normalized modulated GMV images were smoothed with a 4-mm full-width at half-maximum (FWHM) isotropic Gaussian kernel to increase the signal to noise ratio.

For each individual, total GM, WM and CSF were obtained with the Matlab script 'get_totals.m' [33] and used to calculate the individual Total Intracranial Volume (TIV) by summing the volumes of the three already mentioned components (GM, WM, CSF).

## Regional GMV extractions

The WFU Pickatlas [29, 34] was used to generate ROI masks of the selected brain areas in MNI space. Among the different brain areas subdivision generated by WFU Pickatlas, we chose the lobar atlas, and the AAL subdivisions. The lobar ROI subdivisions were as follows: right/left frontal lobe, right/left temporal lobe, right/left parietal lobe, right/left occipital lobe, right/left limbic system, and right/left sublobar area (internal cerebrum: summation of basal ganglia, thalamus, insula, and callosum), right/left brainstem and right/left cerebellum, the AAL subdivision es the 116 area parcellation by Rolls et al. [30]. To automatically extract the

GMV at each ROI for each subject, we programmed a Matlab script based on the MATLAB code "get_totals" [33]. The output of the Matlab script was the regional GMV data for each volunteer at each ROI. Similar or equivalent procedures to extract regional GMV have been used in previous studies [35–37] To verify the truthfulness of the results obtained by the MATLAB "get_totals.m" script, several comparisons were made with the equivalent Marsbar toolbox (available at https://www.nitrc.org/projects/marsbar/). We verified that both tools provided the same results but because "get_totals" was easier to implement inside our script Matlab program we used this method.

## Statistical analysis

Differences in GMV between meditators and controls at each zone/area were analysed by conducting an ad-hoc general linear model (AH-GLM)—ANCOVA that adapts it-self to every area's statistical specificities. The AH-GLM had the following terms Eq (1): the dependent variable (DV) at each area Grey Matter Volume (*GMV*); the factor Meditator (*Med*) with two levels (control Med = 0 and meditator Med = 1); two covariates, the volunteer's age (*Age*) and the volunteer's Total Intracranial Volume (*TIV*); and two interactions, the factor with each covariate: (*Med* × *TIV*) and (*Med* × *Age*) notice that the interactions could be significant only when the associated covariate was significant. At Eq (1) each volunteer is represented by the subscript j and i represents each level of Meditator factor.

$$GMV_{ij} = \beta_0 + Med_i + \beta_1 \cdot Age_{ij} + \beta_2 \cdot TIV_{ij} + \beta_3 \cdot (Med \times TIV)_{ij} + \beta_4 \cdot (Med \times Age)_{ij} + \varepsilon_{ij} \quad (1)$$

Each brain area classification into zones from Zone 1 till Zone 3D was dependent on the statistical significance of each covariates (Age, TIV) and the corresponding interactions (*Med* × *Age*) and (*Med* × *TIV*). Covariates Age and TIV were considered significant at a threshold of *p<0.05*, having a Pearson's correlation coefficient with GMV of *r>0.4*. The interactions (*Med* × *Age*) and (*Med* × *TIV*) were considered significant when their associated covariate was significant and the interaction had *p <0.05*. This way we differentiated zones starting from the simplest Zone 1 where none of the covariates was significant, see Eq (2), to the zone 3D where all covariates and interactions were significant represented by the full model Eq (1).

$$GMV_{ij} = \beta_0 + Med_i + \varepsilon_{ij} \quad (2)$$

Sex was not included into the AH-GLM because one of the conditions to be able to carry out an ANCOVA is that there is no effect of the factors on the covariates that are included in the model. When studying whether there is an effect of sex on the covariate *TIV* it was verified that this effect was highly significant *p < 0.0001*, because males had significant larger *TIV* than females. Therefore, including *TIV* in the model intrinsically controls for the sex factor.

Standardized residuals for the GMV and for the overall model at each zone $\varepsilon_{ij}$ were normally distributed, as assessed by Shapiro-Wilk's test ($p > 0.05$). There was homogeneity of variances, as assessed by visual inspection of a scatterplot and Levene's test of homogeneity of variance ($p <0.05$). There were no outliers in the data, as assessed by no cases with standardized residuals greater than ±3 standard deviations. These models require compliance with two other assumptions: 1. To verify the existence of a non-zero linear relationship between the DV and the covariates in all groups together. If there is no such relationship, conducting an ANCOVA does not make sense, so a unifactorial ANOVA should be conducted alternatively; 2. To check the homogeneity of regression slopes; that is, to ensure that the linear relationship of the DV and the covariate is the same in all groups.

The multiple comparison problem was solved by controlling the false discovery rate (FDR), which manages the expected proportion of false positive findings among all the rejected null

hypotheses [38], by means of the q-values estimated by Storey and Tibshirani's method [39] implemented in neuroscience research by Takeda et al [40]. We should consider that the q value is similar to the p value, with the exception that it is a measure of significance in terms of the false discovery rate rather than the false positive rate. From the distribution of p-values obtained from the multiple comparison, the q-values were provided by means of the Bioconductor´s q-value package from R software (3.6.1, R Foundation for Statistical Computing, Vienna, Austria). Effect size was assessed by means of $\eta^2_{partial}$ following Cohen's benchmarks to define small ($\eta^2_{partial} = 0.01$), medium ($\eta^2_{partial} = 0.06$), and large ($\eta^2_{partial} = 0.14$) effects [39]. Statistical significance was indicated by a false discovery rate (FDR) q-value $<0.05$ or p-value$<0.05$ when corresponds. Further explanation of the AH-GLM and the Zone classification is available at S1 Appendix.

## Results

Our previous study [28] reported two main results: 1. The whole brain had statistically significant larger GMV in meditators compared to controls. 2. There were 5 cluster areas with larger GMV in meditators compared to controls: two from the direct VBM statistical results and three from a priori hypothesised regions with more lenient threshold.

Here we show in Table 2, that the summation of the differences in GMV between meditators versus controls on the above mentioned 5 clusters reflect only around 1.0% of the total GMV difference found at the whole brain: 429.5 mm$^3$ GMV difference in the 5 clusters and 42354.2 mm$^3$ GMV difference in the whole brain, 611.005 (74.633) mm$^3$ in the whole brain GMV in controls versus 653.374 (86.971) mm$^3$ in meditators.

## Lobes area subdivision

In the 16 lobes area subdivision, GMV was statistically significantly larger in meditators compared to non-meditators (FDR q $< 0.05$) in 4 out of 16 areas: R. temporal, R. frontal, R. brainstem and L. frontal. (See Table 3 and Fig 1).

In the two hemispheres GMV was statistically significantly (FDR q $< 0.05$) larger in meditators relative to non-meditators, see Table 4.

The relative GMV difference between meditators and controls showed both extreme cases at brainstem in meditators. On average, the difference in GMV considering all lobes areas was 6.8 ± 3.8% larger in meditators. A similar difference was shown for both hemispheres where the relative difference was always larger GMV for meditators: 7,03% in the right hemisphere

**Table 2. Summary of previous results [28].**

|  | p-value | Vol cluster mm$^3$ | % Diff larger in meditators | Vol diff (Med-Controls) mm$^3$ | $\eta^2_{partial}$ |
|---|---|---|---|---|---|
| R_Insula, vmOFC | 0.023* | 563.6 | 12.6 | 70.8 | 0.194 |
| R_Inf. Temporal, Fusiform Gyrus | 0.037* | 739.1 | 19.6 | 145.2 | 0.300 |
| R_Angular Gyrus | 0.069* | 475.9 | 20.0 | 95.0 | 0.259 |
| L_anterior insula | 0.04 ** | 543.4 | 11.2 | 61.0 | 0.153 |
| L_VLPFC | 0.04 ** | 239.6 | 24.0 | 57.6 | 0.182 |
| Summation of 5 Clusters |  | 2561.6 | 17.5*** | 429.5 |  |
| whole brain | 0.002 | 610961.2 | 6.9 | 42354.2 | 0.207 |

* Non-stationary cluster-level correction based on family wise error.

** A priori hypothesised regions with more lenient threshold.

*** Average of the 5 clusters percentages.

**Table 3. Statistics of GMV differences between groups in the significant lobes (16 areas).**

| Area | Zone * | F | Nom. p-value | FDR q-value | GMV Controls (mean ± std) mL | GMV Medit (mean ± std) mL | **Relat dif % | $\eta^2_{partial}$ |
|---|---|---|---|---|---|---|---|---|
| R. temporal | 3A | 10.52 | 0.002 | 0.016 | 46.65 ± 5.92 | 50.86 ± 7.28 | 9.02 | 0.200 |
| R. frontal | 3A | 10.44 | 0.002 | 0.016 | 78.35 ± 11.61 | 85.68 ± 12.95 | 9.36 | 0.199 |
| R. brainstem | 1 | 9.82 | 0.003 | 0.016 | 1.67 ± 0.28 | 2.00 ± 0.42 | 19.68 | 0.182 |
| L. frontal | 3A | 9.3 | 0.004 | 0.016 | 76.57 ± 11.35 | 83.48 ± 13.4 | 9.02 | 0.181 |

*Further explanation about Zone classification in S1 Appendix.

**Relat dif % = (GMV Medit—GMV Controls) x 100 / GMV Controls.

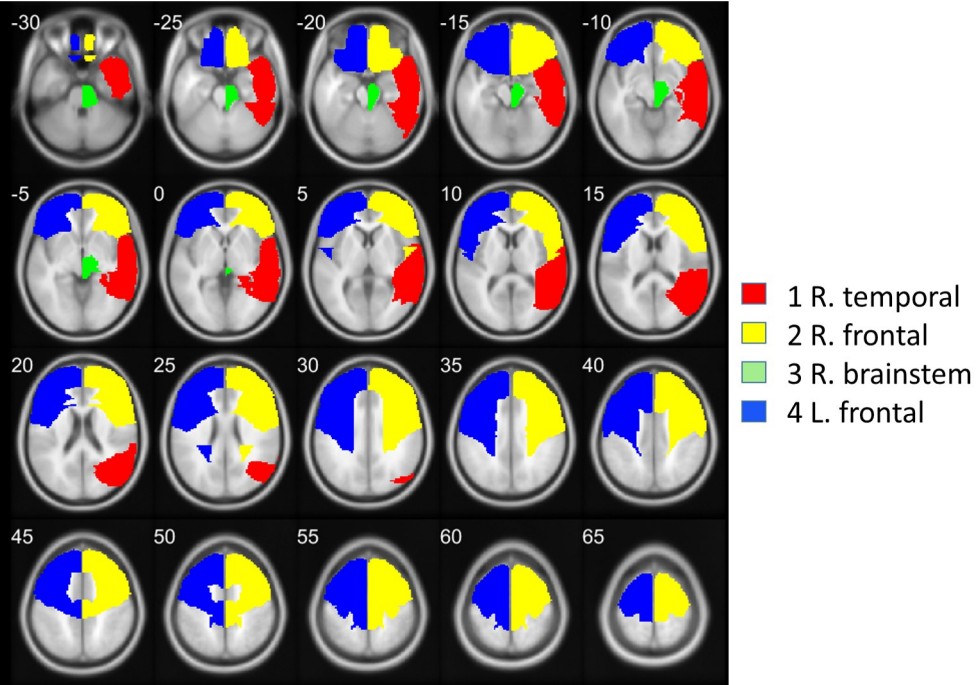

**Fig 1. Axial slices of the lobes area with different GMV between groups, in the order of 1 to 4, following statistical significance.** Z coordinates are shown in mm from the anterior-posterior commissure. The right side of the image corresponds to the right side of the brain.

and 6,72% in the left hemisphere (Table 4). In the whole brain the difference was 6.93%, which was already shown on our previous study [28].

If we consider the reported GMV differences at lobes from Table 3 we see that the summation of the GMV differences in lobes between groups was 20,44 mL or 20440 mm$^3$; this

**Table 4. Statistics of GMV differences between groups in the hemispheres and whole brain.**

| Area | Zone * | F | Nom. p-value | FDR q-value | GMV Controls (mean ± std) mL | GMV Medit (mean ± std) mL | **Relat dif % | $\eta^2_{partial}$ |
|---|---|---|---|---|---|---|---|---|
| R.Hemisph. | 3A | 9.31 | 0.004 | 0.007 | 284.92 ± 35.02 | 304.95 ± 39.76 | 7.03 | 0.182 |
| L.Hemisph. | 3A | 7.94 | 0.007 | 0.007 | 276.62 ± 33.46 | 295.22 ± 39.9 | 6.72 | 0.159 |
| Whole brain GMV | 3A | 9.02 | 0.005 | 0.007 | 611 ± 74.63 | 653.37 ± 86.97 | 6.93 | 0.177 |

*Further explanation about Zone classification in S1 Appendix.

**Relat dif % = (GMV Medit—GMV Controls) x 100 / GMV Controls.

**Table 5. Statistic of GMV differences between groups through significant AAL brain areas.**

| Area | Zone * | F | Nom p- value | FDR q-value | GMV Controls (mean) mm³ | GMV Controls (std) mm³ | GMV Medit (mean) mm³ | GMV Medit (std) mm³ | **Relat dif % | η²partial |
|------|--------|---|--------------|-------------|--------------------------|------------------------|----------------------|---------------------|---------------|-----------|
| MTG.R | 3A | 11.84 | 0.001 | 0.0291 | 14.34 | 1.93 | 15.77 | 2.26 | 9.97 | 0.220 |
| PCL.R | 3A | 11.00 | 0.002 | 0.0291 | 4.15 | 0.40 | 4.32 | 0.52 | 4.10 | 0.208 |
| IFGoperc.R | 3A | 10.47 | 0.002 | 0.0291 | 3.68 | 0.60 | 4.12 | 0.67 | 11.96 | 0.200 |
| PreCG.R | 3A | 9.75 | 0.003 | 0.0291 | 5.92 | 1.06 | 6.75 | 1.23 | 14.02 | 0.188 |
| ITG.R | 3A | 9.30 | 0.004 | 0.0291 | 12.22 | 1.62 | 13.41 | 1.91 | 9.74 | 0.181 |
| IFGorb.R | 3A | 9.08 | 0.004 | 0.0291 | 4.31 | 0.65 | 4.76 | 0.89 | 10.44 | 0.178 |
| PoCG.L | 3A | 8.13 | 0.007 | 0.0382 | 7.70 | 1.25 | 8.42 | 1.22 | 9.35 | 0.162 |
| PreCG.L | 3A | 7.90 | 0.007 | 0.0382 | 7.17 | 1.26 | 7.97 | 1.42 | 11.16 | 0.158 |
| MFG.L | 3A | 7.52 | 0.009 | 0.0393 | 13.34 | 2.07 | 14.57 | 2.34 | 9.22 | 0.152 |
| OLF.L | 3A | 7.45 | 0.009 | 0.0393 | 1.04 | 0.13 | 1.13 | 0.16 | 8.65 | 0.151 |
| MFGorb.R | 3A | 6.88 | 0.012 | 0.0477 | 2.61 | 0.54 | 2.94 | 0.60 | 12.64 | 0.141 |

*Further explanation about Zone classification in S1 Appendix.

**Relat dif % = (GMV Medit—GMV Controls) x 100 / GMV Controls

represent a 48,2% of the total GMV difference reported at the whole brain that was 42354.2 mm³. In the same way the reported GMV difference at the right hemisphere 20,03 mL represents a 47,3% of the whole brain difference while the left hemisphere difference 18.60 mL represents a 43,9%.

## AAL area subdivision

In the 116 AAL area subdivision, GMV was statistically significant (FDR q < 0.05) larger in meditators relative to non-meditators in 11 out of the 116 AAL areas: right middle temporal gyrus (MTG.R), right paracentral lobule (PCL.R), right inferior frontal gyrus opercular part (IFGoperc.R), right precentral gyrus (PreCG.R), right inferior temporal gyrus (ITG.R), right inferior frontal gyrus orbital part (IFGorb.R), left postcentral gyrus (PoCG.L), left precentral gyrus (PreCG.L), left middle frontal gyrus (MFG.L), left olfactory cortex (OLF.L), right middle frontal gyrus orbital part (MFGorb.R), see Table 5 and Fig 2. In 59 AAL areas, the FDR *q*-value was between 0.05 and 0.1.

The GMV difference between meditators and controls ranged from +15.3% larger GMV at Right Parahippocampal gyrus to 0.0%, almost equal, at Right Lenticular nucleus—Pallidum. On average the difference in GMV considering all AAL areas was a 6.7 ± 3.0% larger in Meditators.

If we consider the 11 AAL areas with significant GMV differences, similar to the calculation for the lobe areas, the summation of the difference in GMV between groups on those 11 areas was 6,25 mL which represents a 14.8% of the total GMV difference at the whole brain.

Superposition of AAL and lobes results are shown in Fig 3, while superposition of AAL results and 5 clusters from our first study [28] are shown in Fig 4.

## Discussion

### Discussion of the ad-hoc statistical method

In this study, the statistical analysis made use of an ad-hoc General Linear Model (AH-GLM) which produces more explanatory results for the used subdivisions of the brain.

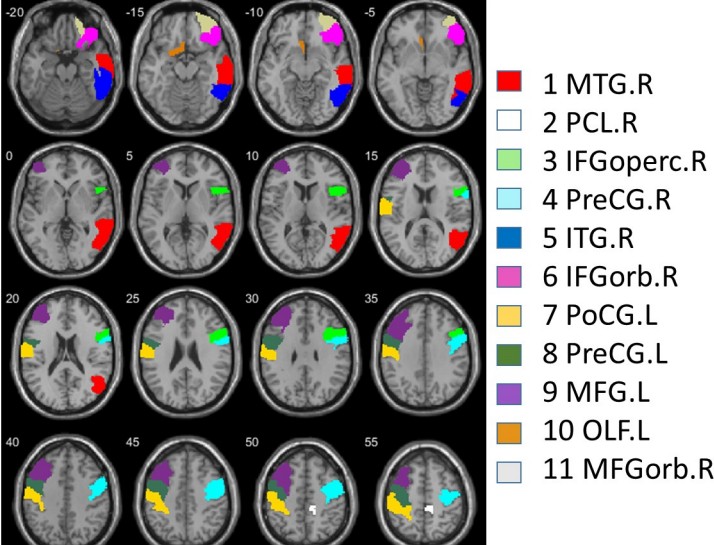

**Fig 2. Horizontal slices of AAL areas with different GMV between groups, in the order of 1 to 11, following statistical significance.** Z coordinates are shown in mm distance from the anterior-posterior commissure. The right side of the image corresponds to the right side of the brain.

As mentioned in the results section, the GMV differences between groups in the 5 clusters reported in our previous study [28] represent only 1% of the total significant GMV difference detected at the whole brain (see Table 2).

Following Cohen's definitions [41], we observed that the effect size was large ($\eta^2_{partial} > 0.14$) in all resulting areas from our AH-GLM (see last columns in Tables 3, 4 and 5), as was the case with the clusters shown in our previous manuscript, (see last column Table 2).

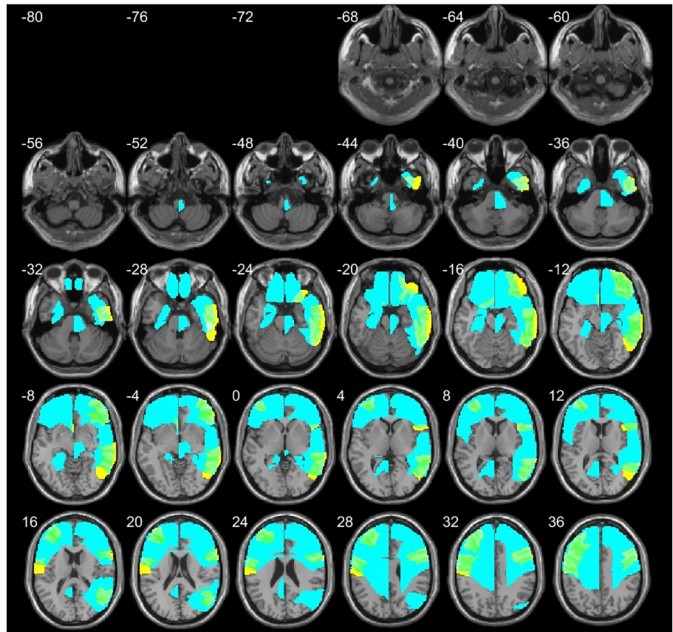

**Fig 3. Lobes results in blue, AAL results in yellow, superposition of both results (AAL an lobes) in green.**

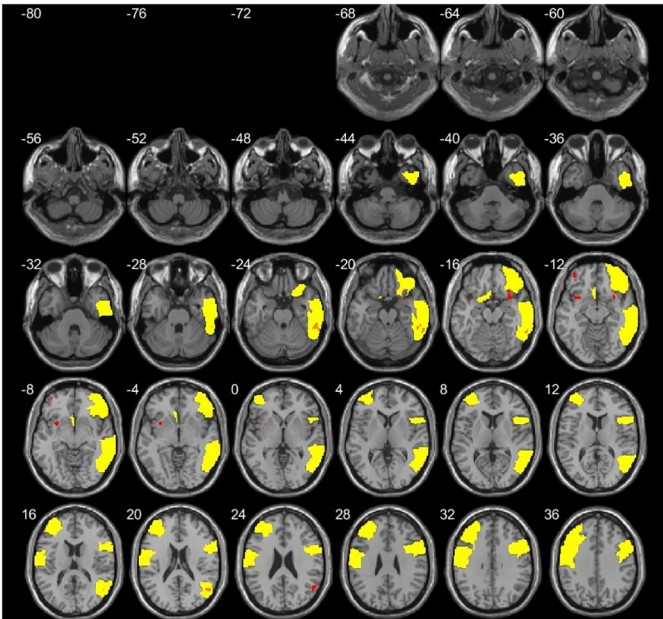

**Fig 4. Shows the results from the AAL areas in yellow and 5 clusters from our first study [28] in red.**

The analysis showed that 11 out of the 116 AAL areas were significantly larger in meditators which represents 14.8% of the total GMV difference at the whole brain (see Table 5). Four out of 16 lobes areas were statistically different in GMV between meditators and non-meditators and represent 20.4% of the GMV differences reported at the whole brain (see Table 3); the left and right hemisphere GMV differences reported represent, respectively, 43.9% and 47,3% of the GMV difference reported at the whole brain (see Table 4).

What these data seem to show is that the larger the number of area subdivisions tested, the smaller the amount of GMV with statistical significance between groups. A possible explanation is the dilution of significant differences at the whole brain with subsequent brain partitions, presumably due to Type II error and to conservative assumptions. This conservative bias may occur in other cross-sectional between-group studies where the whole brain GMV is significantly different between groups, in which case we advocate the use of an AH-GLM like the one presented here. In this situation, we recommend an AH-GLM exhaustive study of the differences between the groups using atlas standard brain subdivisions. The AH-GLM ANCOVA will allow to study in detail the different brain areas of the atlas subdivision used, and this whole process may increase the statistical power of the analyses (see S1 Fig in the S1 Appendix).

Based on our AH-GLM method we present here a more sensitive and detailed examination that reveals significantly different areas that were not detected with the statistical VBM standard procedure. The acknowledgment of these areas will allow to better understand the neuroplastic mechanisms associated with the practice of SYM and its inherent consciousness state of mental silence, discussed in the next section. Our experience is that the standard VBM method based on voxelwise statistics increases spatial specificity as mentioned by Woo et al [42]. On the other hand, region of interest analyses like our AH-GLM may blur significant effects that may extend across large brain regions as mentioned by Poldrack et al. [43]. Due to the important difference in both methods, the results do not overlap (see Fig 4) so we think that the results from both methods complement each other. Importantly, our results support the

common approach of regressing out TIV and age effects in VBM analysis. Nonetheless, our results may suggest that the appropriateness of regressing out TIV and age effects are only necessary at cortical ROI, but this was not the case at the brain stem (see S1 Appendix). A post-hoc explanation is to consider that TIV is not related to regional volumes of brain stem regions (e.g. larger brains do not have larger brain stems) and, therefore, there is no need to correct for TIV (either age) effects on brain stem but on cortical brain regions. However, previous reports have shown consistent TIV (but partial age) linear association with brain stem structures [44, 45] using different methodological approaches. Therefore, our results on ROI analyses support the common ANCOVA design regressing out TIV (e.g. for estimation of regional brain volumes) and age (e.g., usually indirectly related to regional brain volume across the brain) in cortical brain regions. Moreover, further studies may explore whether AH-GLM ANCOVA better adapts it-self to every area's statistical specificities rather than other more common problematic approaches based on voxelwise or clusterwise thresholds [46] at the cost of spatial resolution. In this sense, multi-modal parcellation of brain regions, merged within the context of the Human Connectome, project may improve the definition of ROIs analyses with larger sample sizes [47].

## Discussion of the VBM results

The 3 lobe areas with the largest significant GMV differences were in the right hemisphere: right temporal, right frontal and right brainstem. Furthermore, the 6 AAL areas with the largest significant GMV differences were also in the right hemisphere: in middle and inferior temporal lobe, in inferior and orbital frontal cortices, and in para- and precentral lobes (Tables 3 and 5, Figs 1 and 2).

Compared to our previous study [28] where we found GMV differences in insula/vmOFC, inferior temporal and parietal lobes, with our new lobe and AAL area analyses, we detected additional brain regions to be different between Meditators and Non-Meditators, including inferior and orbitofrontal cortex, middle temporal lobe, precentral and paracentral gyrus and brainstem while replicating findings in OFC and inferior temporal lobe. Below we discuss mainly the findings in the novel areas detected.

This prevalence of larger differences in GMV in areas of the right hemisphere is in concordance with our previous publications of functional and structural MRI associated with the long-term practice of SYM [28, 48] where we found increased neuronal activation of right hemispheric regions of right inferior frontal cortex and superior temporal lobe in long-term SYM during their meditation relative to rest and significantly larger GMV in areas mainly of the right hemisphere in anterior insula, inferior temporal gyrus and angular gyrus. It is also in line with a study that tested only 4 weeks of SYM training and found an enlargement in right inferior frontal cortex in the Meditation training group compared to controls [49].

The most significant AAL area shown in Table 5 was the right middle temporal gyrus (MTG.R). Larger GMV in this region has been associated with feelings of "intimate relationship with God and engaging in religious behaviour" in different religious practices [50]. Furthermore, MTG.R has been shown to be activated in Carmelite nuns in relation to the subjective impression of deepening into the spiritual dimension [51]. SYM practitioners also mention that through their meditation they have subjective experiences of spirituality and union with the divine (yoga = union).

The middle and inferior frontal lobes are crucial for higher order executive functions and emotion control [52, 53]. The middle frontal lobes, also in the AAL area subdivision analysis, are crucial for top-down emotion control, as well as for working memory, planning and other executive functions [54].

The inferior frontal lobes, which were observed also in the AAL area subdivision analysis, are crucial for executive functions such as sustained attention, working memory, performance monitoring, switching and inhibitory self-control [55]. The finding of larger GMV in these regions is in line with previous VBM studies of other meditation techniques that also found larger frontal lobe volumes in long-term Meditators, in particular in inferior frontal regions [56]. The findings suggest that long-term meditation leads to enlargement of inferior frontal lobe regions possibly due to the fact that meditation which teaches the practitioner to inhibit unwanted thoughts and control their attention is a powerful attention and self-control training which may lead to the enlargement of areas that mediate attention and inhibitory self-control [57–60]. This would be in line with several studies that have shown that long-term Meditators have better performance in tasks of executive functions, in particular in tasks of sustained attention and inhibitory self-control [2, 61, 62]. Meditation, however, also has shown to lead to better emotional detachment [63] and emotional self-control which is mediated by the orbito-frontal and ventromedial frontal regions [53]. In fact, the orbitofrontal cortex was already been shown to be enlarged in our previous more stringent VBM analysis of these data [64].

The enlargement in the temporal lobe is also interesting. The middle and inferior right temporal lobes are closely connected to the limbic system and form crucial part of the emotion control network [54–56]. The middle and inferior temporal gyri were also found to be enhanced in activation in a meta-analysis of fMRI studies [65].

The enlargement in the brainstem is of particular interest, as previous studies have found increased GMV in long-term meditators relative to controls in the brainstem [66, 67]; in a longitudinal study of mindfulness meditation, this increase of GMV in the brainstem in the meditators was associated with better well-being [68]. The brainstem contains several production areas of several modulatory neurotransmitter pathways, such as those arising from the raphe nuclei (serotonergic; associated with modulation of mood and cognitive functions), ventral tegmental area (dopaminergic; associated with motivation, working memory and attention) and locus coeruleus (noradrenergic; associated with arousal and attention) [68, 69]. The state of mental silence has been described subjectively in meditation scriptures as a state of enhanced alertness, attention and arousal [1, 2].

The autonomic nervous system, brainstem and cortical systems are closely interconnected in their mediation of the regulation of behaviour and cognition [70]. The enlargement of the brainstem in long-term Meditators is therefore potentially a consequence of the long-term practice of achieving the state of thoughtless awareness which leads to enhanced alertness and arousal. It may also be related to the activation of the autonomic nervous system during meditation [71] that is closely interconnected with brainstem regions. Given that the brainstem is closely interconnected with frontal regions. It is also of note that brainstem and the two frontal lobes were also increased in GMV in long-term Meditators.

The 6,9% larger GMV in meditators at the whole brain with a p-value of 0.002 constitutes as far as we know the largest difference in GMV between healthy groups of similar age and conditions. No other meditation technique or practice has shown such a large statistical difference in GMV at the whole brain. One of the assumptions of SYM is the spontaneous (Sahaja = spontaneous) awakening of the Kundalini energy [72] during the meditation which allows the practitioners to perceive the achievement of yoga (yoga = union) and the state of mental silence, which meditators subjectively perceive as a cool breeze when they put their hands some centimetres above of their head. It is possible that this experience, which is specific to SYM, may be related to the enlargement of VBM and this needs to be further tested.

The pre, post and paracentral gyri were also different in Meditators. A meta-analysis of structural and fMRI studies found precentral gyrus to be increased in volume and activation in relation to meditation [65]. The precentral gyrus has furthermore been found to be larger in

cortical gyrification in long-term Meditators [73] and increased in the integrity of a fiber tract predominantly originating/terminating in motor areas (cortical-spinal tract) and been shown to have larger GM tissue in paracentral regions [74, 75]. fMRI studies on interoceptive awareness showed enhanced activity of the lateral somatomotor cortex among other regions such as the insula [76]. It is thus possible that regions in the vicinity of the motor cortices, perhaps in their associations with the insula might aid in mediating interoceptive attention and awareness which is typically enhanced in Meditation [73]. There is furthermore evidence that meditation leads to reduced pain sensitivity which is mediated by sensorimotor regions [77].

One important limitation of this research, which is inherent to all cross-sectional grey matter studies on group differences, is that we cannot assure that the only possible cause of the group differences are differences in the behaviour tested, i.e., meditation in this case. It is possible that the GMV differences are attributable to other mediating factors such as personality, lifestyle, etc. One possible way to address this confound would be to conduct longitudinal randomised controlled trials where GMV is tested before and after several months of meditation practice. One such longitudinal RCT, however, did find increased right frontal GMV after 4 weeks of SYM meditation practice [49], corroborating at least part of the here observed findings.

## Conclusions

In our previous study [28] where we used the standard statistical model for VBM, only 5 relatively small brain areas were statistically different in GMV between groups. These 5 areas represented only around 1% of the total 6.9% larger GMV difference shown at the whole brain in meditators compared to non-meditators. Hence the possibility of a Type II error or conservative results was considered. In this study, using an ad-hoc statistical method, we show in more detail how this 6,9% larger GMV in meditators, the largest GMV difference in healthy groups of similar age and conditions in the literature so far, is distributed in the brain subregions of the meditators. The larger GMV in meditators is focused in particular in the right hemisphere in frontal and temporal brain areas related to attention and emotional control.

## Supporting information

**S1 Appendix. Further explanation of the ad-hoc statistical model.**
(DOCX)

**S1 Table. Lobes GMV data of healthy controls and meditators.**
(XLSX)

**S2 Table. 116 AAL GMV data of healthy controls and meditators.**
(XLSX)

**S3 Table. Statistics of GMV differences between groups in the 16 lobes areas.**
(XLSX)

**S4 Table. Statistic of GMV differences between groups in the 116 AAL brain areas.**
(XLSX)

## Acknowledgments

We acknowledge the technical and logistical support of MRI services for Biomedical Studies (Servicio de Resonancia Magnética para Investigaciones Biomédicas) of the University of La Laguna.

## Author Contributions

**Conceptualization:** Sergio Elías Hernández, José Suero, Alfonso Barros-Loscertales, José Luis González-Mora, Katya Rubia.

**Data curation:** Sergio Elías Hernández.

**Formal analysis:** Sergio Elías Hernández, Roberto Dorta, Alfonso Barros-Loscertales.

**Funding acquisition:** Sergio Elías Hernández.

**Investigation:** Sergio Elías Hernández, Roberto Dorta, José Suero, Alfonso Barros-Loscertales, José Luis González-Mora, Katya Rubia.

**Methodology:** Sergio Elías Hernández, Roberto Dorta, José Suero, Alfonso Barros-Loscertales, Katya Rubia.

**Project administration:** Sergio Elías Hernández.

**Resources:** Sergio Elías Hernández, Alfonso Barros-Loscertales, Katya Rubia.

**Software:** Sergio Elías Hernández.

**Supervision:** Sergio Elías Hernández, José Suero, Alfonso Barros-Loscertales, José Luis González-lez-Mora, Katya Rubia.

**Validation:** Sergio Elías Hernández, Roberto Dorta, Alfonso Barros-Loscertales, Katya Rubia.

**Visualization:** Sergio Elías Hernández, Alfonso Barros-Loscertales, Katya Rubia.

**Writing – original draft:** Sergio Elías Hernández, Roberto Dorta, Alfonso Barros-Loscertales, Katya Rubia.

**Writing – review & editing:** Sergio Elías Hernández, Roberto Dorta, José Suero, Alfonso Barros-Loscertales, José Luis González-Mora, Katya Rubia.

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
