## [Decision Letter · Decision Letter 0]

30 Sep 2020

PONE-D-20-23078

Increases in whole brain grey matter associated with long-term Sahaja Yoga Meditation: a detailed area by area description

PLOS ONE

Dear Dr. Hernández,

Thank you for submitting your manuscript to PLOS ONE. After careful consideration, we feel that it has merit but does not fully meet PLOS ONE’s publication criteria as it currently stands. Therefore, we invite you to submit a revised version of the manuscript that addresses the points raised during the review process.

Beyond the recommendations of the Reviewers, I would also like to point out a couple of areas that should be addressed as well:

1. At times, there seems to be too much weight given to p-values (e.g. "ten times more significant"). I encourage you to assess effect sizes rather than relying solely on p-values.

2. Please us the term "sex" rather than "gender". Sex carries the biological meaning whereas gender is the social construct, which is not relevant in the context of your manuscript.

3. In the description of the VBM analysis, you state that step 5 utilized affine normalization. This seems highly unlikely as the registration should almost certainly be nonlinear. Please clarify.

4. It is unclear what echo-planar image was used in this study. Please clarify.

5. Random field theory is not the only method that can be used, as stated in the Introduction. Permutation-based methods are also used. In fact, I would encourage you to also use a permutation-based technique to assess whether your results can be replicated. 

We look forward to receiving your revised manuscript.

Kind regards,

Niels Bergsland

Academic Editor

PLOS ONE

Journal Requirements:

2.We note that you have indicated that data from this study are available upon request. PLOS only allows data to be available upon request if there are legal or ethical restrictions on sharing data publicly. For more information on unacceptable data access restrictions, please see http://journals.plos.org/plosone/s/data-availability#loc-unacceptable-data-access-restrictions.

Reviewers' comments:

Reviewer's Responses to Questions

**Comments to the Author**

1. Is the manuscript technically sound, and do the data support the conclusions?

Reviewer #1: Yes

Reviewer #2: Partly

2. Has the statistical analysis been performed appropriately and rigorously? 

Reviewer #1: Yes

Reviewer #2: I Don't Know

3. Have the authors made all data underlying the findings in their manuscript fully available?

Reviewer #1: No

Reviewer #2: No

4. Is the manuscript presented in an intelligible fashion and written in standard English?

Reviewer #1: Yes

Reviewer #2: Yes

5. Review Comments to the Author

Reviewer #1: Current manuscript by Hernandez et al., entitled “Increases in whole brain grey matter associated with long-term Sahaja Yoga Meditation: a detailed area by area description” investigates gray matter volume differences between long-term meditation practice and controls. The authors use the same dataset from a previous publication and report that their previous findings (5 clusters) reflect only around 1.0% of the total GMV difference. They investigate gray matter volume changes using an atlas based approach using two different atlases, namely 16 lobar areas and AAL atlas 116 subdivisions.

First, the rationale for using two different subdivisions is not clear – e.g. did they expect an overlap or did they use the results of the first atlas as a way to inform the second?

The lobar approach seems too broad and not an important contribution to existing research in the field.

The AAL atlas produces some interesting results, which seems to be overlapping with previous literature. Yet, this brings the novelty aspect into question. The discussion focuses broadly on various brain lobes and on their function and any novel findings -if existent- is not emphasized.

Second, if the purpose of the paper is to advocate the use of an atlas-based approach instead of a whole-brain GMV approach, this then needs to be discussed in further detail, including the rationale for their attempt to increase statistical power.

Also, It seems like there is very little overlap between findings from their whole-brain approach and ROI approach. It would be interesting to see both AAL ROI results and whole brain approach mapped onto same brain. If there is very little overlap, this needs to be further discussed.

It’s very important not to draw any conclusions from LTM data with respect to how meditation changes the brain. The GMV differences may be attributable to other factors such as personality, life style etc. I recommend that the authors acknowledge this as a limitation.

I suggest the authors avoid terms such as “which is felt like a cool breeze of energy on top of the head”, and use terms from cognitive science to appeal to scientific audience.

Overall, it is an interesting paper that may contribute to the neuroscience of meditation.

Reviewer #2: Please comment on your technique relative to this articel

PNAS

https://www.pnas.org/content/113/28/7900

Cluster failure: Why fMRI inferences for spatial extent have inflated false-positive rates

Anders Eklunda,b,c,1, Thomas E. Nicholsd,e, and Hans Knutssona,c

"These findings speak to the need of validating the statistical methods being used in the field of neuroimaging.

6. PLOS authors have the option to publish the peer review history of their article (what does this mean?). If published, this will include your full peer review and any attached files.

Reviewer #1: No

Reviewer #2: No

---

## [Author Response · Author response to Decision Letter 0]

14 Nov 2020

Dear Dr. Niels Bergsland

We are very thankful for giving us the opportunity to revise and resubmit our manuscript to Plos One. As you will see below, we have addressed all the editor’s and the reviewers’ concerns in the new version. 

We are also thankful for all the valuable comments received that allowed us to improve our manuscript. We hope that you will conclude that the current version of the manuscript meets the high standard of the journal.

In order to facilitate the revision, we used in this document highlight colours for our responses: grey highlight colour for responses to the Referees and the Editor, and yellow highlight colour in the text that, in addition to responding to Referees or the Editor, is also included in the new version of our manuscript. The original text from the Editor, the Referees or the Journal is in bold italic type without highlight colour.

We respectfully look forward to your editorial decision.

Best regards,

Sergio Elías Hernández

On behalf of all the co-authors

University of La Laguna

Responses to the editor ‘s and the reviewers' comments:

Editor: 

1. At times, there seems to be too much weight given to p-values (e.g. "ten times more significant"). I encourage you to assess effect sizes rather than relying solely on p-values.

Response Nº1: Following the Editor’s advice, we have introduced the effect sizes, they have been added in the last columns in tables 2, 3, 4 and 5. We have deleted the paragraph with the sentence "ten times more significant" and we have written two new paragraphs:

1. In the last paragraph of the “Statistical Analysis” subsection, we have added:

Effect size was assessed by means of η_(partial )^2 following Cohen’s benchmarks to define small (η_(partial )^2 = 0.01), medium (η_(partial )^2 = 0.06), and large (η_(partial )^2 = 0.14) effects [39].

2. In the “Discussion of the ad-hoc statistical method” subsection, we have added:

Following Cohen’s definitions [41] we could see that the effect size was large (η_(partial )^2 > 0.14) in all resulting areas from our ad-hoc statistical model (see last columns in Tables 5, 4 and 3), as was the case with the clusters shown in our previous manuscript, (see last column Table 2).

Here we should mention that we have omitted in the new version of our manuscript the trend level result shown in the previous version in Table 3 (Lobes areas) at L. Limbic because the effect size at L. limbic was medium and not large as the rest of results in Tables 2, 3, 4 and 5.

2. Please us the term "sex" rather than "gender". Sex carries the biological meaning whereas gender is the social construct, which is not relevant in the context of your manuscript.

Response Nº2: Following the editor’s advice in the new version of our manuscript the term “gender” has been replaced with “sex”.

3. In the description of the VBM analysis, you state that step 5 utilized affine normalization. This seems highly unlikely as the registration should almost certainly be nonlinear. Please clarify. 

Response Nº3: Certainly, MRI anatomical images involve linear and nonlinear deformations during DARTEL based normalization technique, although the final DARTEL to MNI registration involves just affine transformations. 

For the preprocessing of MRI anatomical images we followed the steps described by John Ashburner, one of the authors of the VBM method, see references 21-24 in our manuscript. In John Ashburner’s document “VBM Tutorial” (https://www.fil.ion.ucl.ac.uk/~john/misc/VBMclass10.pdf) on page 8 it states: “This template is registered to MNI space (affine transform), allowing the transformations to be combined so that all the individual spatially normalised scans can also be brought into MNI space”. In order to avoid confusion, now at step 5 in the Voxel-Based Morphometry subsection it is written:

5. The final DARTEL template image modulated in the previous step was registered to MNI space so that all the individual spatially normalised scans were brought into MNI space. 

4. It is unclear what echo-planar image was used in this study. Please clarify. 

Response Nº4: Related with this point, we have found a mistake so now in the “MRI Acquisition” subsection one can read: 

High resolution sagittally oriented anatomical images were collected. A 3D fast spoiled-gradient recalled pulse sequence was obtained (TR = 8.8 ms, TE = 1.7 ms, flip angle = 10º, matrix size = 256 × 256 pixels, 1 × 1 mm in plane resolution, spacing between slices = 1 mm plus 0 mm interslice gap, slice thickness = 1 mm).

5. Random field theory (RFT) is not the only method that can be used, as stated in the Introduction. Permutation-based methods are also used. In fact, I would encourage you to also use a permutation-based technique to assess whether your results can be replicated. 

Response Nº 5: We agree with the Editor that the RFT thresholding is not the only method used in SPM´s standard inference. Permutation-based methods can also be used without relying on distribution assumptions and offering the possibility of correcting for family wise error. According to Eklund et al. (2012, 2016, 2019) random permutation-based analyses techniques are more suitable for (f)MRI data than parametric testing (Adolf et al., 2014; Front NeuroInform). Similarly, Woo et al., (2014; Neuroimage) changed the trend in reporting cluster threshold correction from a more stringent voxelwise thresholding. Parametric tests on MRI signal problems are related with temporal time series, but are not so much of a problem in structural MRI datasets as far as we know. In fact, similar results have been shown when applying both thresholding techniques to VBM datasets (Shiino et al., 2017; Sci Rep). 

We therefore consider our approach as appropriate, based on the assumptions of our AH-GLM ANCOVA analysis of variance. We have further commented on this issue in Response 14. See below in the “Discussion of the ad-hoc statistical method” we have added:

Moreover, further studies may explore whether AH-GLM ANCOVA better adapts it-self to every area’s statistical specificities rather than other more common problematic approaches based on voxelwise or clusterwise thresholds [46] at the cost of spatial resolution. In this sense, multi-modal parcellation of brain regions, merged within the context of the Human Connectome, project may improve the definition of ROI analyses with larger sample sizes [47].

JOURNAL REQUIREMENTS:

2.We note that you have indicated that data from this study are available upon request. PLOS only allows data to be available upon request if there are legal or ethical restrictions on sharing data publicly. For more information on unacceptable data access restrictions, please see http://journals.plos.org/plosone/s/data-availability#loc-unacceptable-data-access-restrictions.

Response Nº6: In this new version of our manuscript we have included 2 supplementary excel files with anonymized data sets necessary to replicate our study findings: 

S1 Table. Lobes Data detailed information of healthy controls and meditators .xlsx

S2 Table. 116 AAL Data detailed information of healthy controls and meditators .xlsx

So now on Data Availability, it can be read:

Data Availability: All relevant data are within the manuscript and its Supporting Information files.

5. Review Comments to the Author

Reviewer #1: Current manuscript by Hernandez et al., entitled “Increases in whole brain grey matter associated with long-term Sahaja Yoga Meditation: a detailed area by area description” investigates gray matter volume differences between long-term meditation practice and controls. The authors use the same dataset from a previous publication and report that their previous findings (5 clusters) reflect only around 1.0% of the total GMV difference. They investigate gray matter volume changes using an atlas based approach using two different atlases, namely 16 lobar areas and AAL atlas 116 subdivisions.

1a. First, the rationale for using two different subdivisions is not clear – e.g. did they expect an overlap or did they use the results of the first atlas as a way to inform the second?.

Response Nº8: To answer this question we have introduced the following texts and figures in the Introduction section:

We used these two brain atlas subdivisions because they are widely used in the neuroscience literature and because the proposed methodology allows us to study both of them in detail. This way we could test the application of our ad-hoc method in 2 different scenarios and observe if the results obtained provided some overlap and coherence.

In the Results section, the following text and the new figure 3 have been added:

Fig 3. Lobes results in blue, AAL results in yellow, superposition of both results (AAL an lobes) in green

1b. The AAL atlas produces some interesting results, which seems to be overlapping with previous literature. Yet, this brings the novelty aspect into question. The discussion focuses broadly on various brain lobes and on their function and any novel findings -if existent- is not emphasized.

Response Nº9: We would like to thank the Reviewer for pointing out the need to emphasize the contribution of our study. We have explicitly stated these contributions in the “Discussion of the ad-hoc statistical method” subsection of the manuscript:

Importantly, our results support the common approach of regressing out TIV and age effects in VBM analysis. Nonetheless, our results may suggest that the appropriateness of regressing out TIV and age effects are only necessary at cortical ROIs, but this was not the case at the brain stem (see supplementary material S1 Appendix). A post-hoc explanation is to consider that TIV is not related to regional volumes of brain stem regions (e.g. larger brains do not have larger brain stems) and, therefore, there is no need to correct for TIV (either age) effects on brain stem but on cortical brain regions. However, previous reports have shown consistent TIV (but partial age) linear association with brain stem structures [44, 45] using different methodological approaches. Therefore, our results on ROI analyses support the common ANCOVA design regressing out TIV (e.g. for estimation of regional brain volumes) and age (e.g., usually indirectly related to regional brain volume across the brain) in cortical brain regions.

To emphasize the AAL atlas results we have added in the “Discussion of the VBM results” subsection the following paragraphs:

The most significant AAL area shown in table 5 was the right middle temporal gyrus (MTG.R). Larger GMV in this region has been associated with feelings of “intimate relationship with God and engaging in religious behaviour” in different religious practices [50]. Furthermore, MTG.R has been shown to be activated in Carmelite nuns in relation to the subjective impression of deepening into the spiritual dimension [51]. SYM practitioners also mention that through their meditation they have subjective experiences of spirituality and union with the divine (yoga = union).

The middle frontal lobes, also in the AAL area subdivision analysis, are crucial for top-down emotion control, as well as for working memory, planning and other executive functions [54].

The middle and inferior right temporal lobes are closely connected to the limbic system and form crucial part of the emotion control network [54-56]. The middle and inferior temporal gyri were also found to be enhanced in activation in a meta-analysis of fMRI studies [65]

The pre, post and paracentral gyri were also different in Meditators. A meta-analysis of structural and fMRI studies found precentral gyrus to be increased in volume and activation in relation to meditation [65]. The precentral gyrus has furthermore been found to be larger in cortical gyrification in long-term Meditators [73] and increased in the integrity of a fiber tract predominantly originating/terminating in motor areas (cortical-spinal tract) and been shown to have larger GM tissue in paracentral regions [74, 75]. fMRI studies on interoceptive awareness showed enhanced activity of the lateral somatomotor cortex among other regions such as the insula [76]. It is thus possible that regions in the vicinity of the motor cortices, perhaps in their associations with the insula might aid in mediating interoceptive attention and awareness which is typically enhanced in Meditation [73]. There is furthermore evidence that meditation leads to reduced pain sensitivity which is mediated by sensorimotor regions [77].

2.-Second, if the purpose of the paper is to advocate the use of an atlas-based approach instead of a whole-brain GMV approach, this then needs to be discussed in further detail, including the rationale for their attempt to increase statistical power. 

Response Nº10: To deal with this second question we have added to the manuscript in the “Discussion of the ad-hoc statistical method” subsection the following 2 paragraph:

In this study, the statistical analysis made use of an ad-hoc ANCOVA General Linear Model which produces more explanatory results for the used subdivisions of the brain. 

This conservative bias may occur in other cross-sectional between-group studies where the whole brain GMV is significantly different between groups, in which case we advocate the use of an ad-hoc GLM method like the one presented here. In this situation, we recommend an ad-hoc exhaustive study of the differences between the groups using atlas standard brain subdivisions. The ad-hoc ANCOVA model will allow to study in detail the different brain areas of the atlas subdivision used, and this whole process may increase the statistical power of the analyses (see Fig 1S in the S1 Appendix).

3.- Also, it seems like there is very little overlap between findings from their whole-brain approach and ROI approach. It would be interesting to see both AAL ROI results and whole brain approach mapped onto same brain. If there is very little overlap, this needs to be further discussed.

Response Nº11: We agree with the reviewer’s suggestion and we would like to thank his/her idea to introduce a picture showing the overlap between both studies. We have pictured such overlap in the new Fig 4, at the Results section and commented in the Results and the Discussion sections.

In the Results section:

Fig 4. Shows the results from the AAL areas in yellow and 5 clusters from our first study [28] in red.

In the “Discussion of the ad-hoc statistical method” section:

Our experience is that the standard VBM method based on voxelwise statistics increases spatial specificity as mentioned by Woo et al [42]. On the other hand, region of interest analyses like our ad-hoc method may blur significant effects that may extend across large brain regions as mentioned by Poldrack et al. [43]. Due to the important difference in both methods, the results do not overlap (see Fig 4) so we think that the results from both methods complement each other

In the “Discussion of the VBM results” subsection:

Moreover, further studies may explore whether AH-GLM ANCOVA better adapts it-self to every area’s statistical specificities rather than other more common problematic approaches based on voxelwise or clusterwise thresholds [46] at the cost of spatial resolution. In this sense, multi-modal parcellation of brain regions, merged within the context of the Human Connectome, project may improve the definition of ROIs analyses with larger sample sizes [47].

4.- It’s very important not to draw any conclusions from LTM data with respect to how meditation changes the brain. The GMV differences may be attributable to other factors such as personality, life style etc. I recommend that the authors acknowledge this as a limitation. 

Response Nº12: Thank you, we totally agree with this statement so we have added in the manuscript the following text in the Discussion section:

One important limitation of this research, which is inherent to all cross-sectional grey matter studies on group differences, is that we cannot assure that the only possible cause of the group differences are differences in the behaviour tested, i.e., meditation in this case. It is possible that the GMV differences are attributable to other mediating factors such as personality, lifestyle, etc. One possible way to address this confound would be to conduct longitudinal randomised controlled trials where GMV is tested before and after several months of meditation practice. One such longitudinal RCT, however, did find increased right frontal GMV after 4 weeks of SYM meditation practice [49], corroborating at least part of the here observed findings.

5.- I suggest the authors avoid terms such as “which is felt like a cool breeze of energy on top of the head”, and use terms from cognitive science to appeal to scientific audience.

 Response Nº13: We followed the referee’s suggestion and we have omitted the term “energy” that could be controversial so the new sentence can be read: 

which meditators subjectively perceive as a cool breeze when they put their hands some centimetres above of their head.

Reviewer #2: Please comment on your technique relative to this article

PNAS https://www.pnas.org/content/113/28/7900

Cluster failure: Why fMRI inferences for spatial extent have inflated false-positive rates

Anders Eklunda,b,c,1, Thomas E. Nicholsd,e, and Hans Knutssona,c

"These findings speak to the need of validating the statistical methods being used in the field of neuroimaging. 

Response Nº14: Eklund et al. (2016; 2012; 2019) comment on several inaccuracies in parametric methods that lead to high rates of false positives. However, some differences between our and these studies make them not comparable. First, these studies refer to fMRI data sets rather than structural MRI. Second, the main reason for false positives is related to the application of clusterwise inference. In our study, we do not apply either voxelwise or clusterwise inference but ROI analysis, which offers a control for Type I error by limiting the number of statistical tests. Nonetheless, it may be discussed that ROI analysis may approach a clusterwise inference, since a ROI involves an anatomical extension rather than the cluster extension after a voxelwise threshold is set. But even after trying to visualize these resemblances, two main differences may be noticed between our ROI analysis and clusterwise thresholds. First, our ROI analysis is not restricted to significant voxels for statistical contrast in order to extract statistical inferences. On the contrary, the statistical inference would be biased (i.e. circularity bias). Likewise, we applied ROI analysis without requiring all voxels to be above a statistical threshold, but simply averaging across the entire region. In this sense, the application of Gaussian random-field theory for FWE-corrected voxelwise and clusterwise are to be considered different in our study. Nonetheless, the assumption that affects clusterwise inference in the context of RFT applied to structural MRI is that spatial smoothness is not constant over the brain (Hayasaka et al., 2004). In other words, there might be significant effects with different extent thresholds depending on the smoothing effects and region extent. In our analyses, we did not apply voxelwise or clusterwise thresholds but ROI values were extracted from regions which may be differently affected by the smoothing (e.g., pallidum). In fact, spatially smoothing data satisfies the assumption that the residuals follow a Gaussian distribution. Finally, we applied FDR as a way of correcting for family-wise error which is not commented in Eklund et al., 2016.

In sum, we expect that the reviewer agrees that our analysis is not comparable to the ones discussed in Eklund et al., 2016. Related with this at the end of the “Discussion of the ad-hoc statistical method” subsection we have added:

Moreover, further studies may explore whether AH-GLM ANCOVA better adapts it-self to every area’s statistical specificities rather than other more common problematic approaches based on voxelwise or clusterwise thresholds [46] at the cost of spatial resolution. In this sense, multi-modal parcellation of brain regions merged within the context of the Human Connectome project may improve the definition of ROIs analyses with larger sample sizes [47].

Associated with this manuscript is the S1 Appendix that contains detailed explanation of the AH-GLM implementation.

On this new version of our manuscript, we have added the following supporting information:

S1 Appendix. Further explanation of the ad-hoc statistical model.

S1 Table. Lobes GMV Data of healthy controls and meditators

S2 Table. 116 AAL GMV Data of healthy controls and meditators

S3 Table. Statistics of GMV differences between groups in the 16 lobes areas

S4 Table. Statistic of GMV differences between groups in the 116 AAL brain areas

---

## [Decision Letter · Decision Letter 1]

7 Dec 2020

Increases in whole brain grey matter associated with long-term Sahaja Yoga Meditation: a detailed area by area description

PONE-D-20-23078R1

Dear Dr. Hernández,

We’re pleased to inform you that your manuscript has been judged scientifically suitable for publication and will be formally accepted for publication once it meets all outstanding technical requirements.

Kind regards,

Niels Bergsland

Academic Editor

PLOS ONE

Additional Editor Comments (optional):

Reviewers' comments:

Reviewer's Responses to Questions

**Comments to the Author**

1. If the authors have adequately addressed your comments raised in a previous round of review and you feel that this manuscript is now acceptable for publication, you may indicate that here to bypass the “Comments to the Author” section, enter your conflict of interest statement in the “Confidential to Editor” section, and submit your "Accept" recommendation.

Reviewer #1: All comments have been addressed

2. Is the manuscript technically sound, and do the data support the conclusions?

Reviewer #1: Yes

3. Has the statistical analysis been performed appropriately and rigorously? 

Reviewer #1: Yes

4. Have the authors made all data underlying the findings in their manuscript fully available?

Reviewer #1: Yes

5. Is the manuscript presented in an intelligible fashion and written in standard English?

Reviewer #1: Yes

6. Review Comments to the Author

Reviewer #1: Thank you for addressing all of my comments. I'd like to recommend a change in the title as the current version is slightly misleading.

7. PLOS authors have the option to publish the peer review history of their article (what does this mean?). If published, this will include your full peer review and any attached files.

Reviewer #1: **Yes: **Gunes Sevinc

---

## [Editor Report · Acceptance letter]

14 Dec 2020

PONE-D-20-23078R1 

Larger whole brain grey matter associated with long-term Sahaja Yoga Meditation: a detailed area by area comparison 

Dear Dr. Hernández:

I'm pleased to inform you that your manuscript has been deemed suitable for publication in PLOS ONE. Congratulations! Your manuscript is now with our production department. 

Kind regards, 

on behalf of

Dr. Niels Bergsland 

Academic Editor

PLOS ONE